# Pointer Sentinel Mixture Models

**Stephen Merity, Caiming Xiong, James Bradbury & Richard Socher**
MetaMind - A Salesforce Company
Palo Alto, CA, USA
`{smerity,cxiong,james.bradbury,rsocher}@salesforce.com`

## ABSTRACT

Recent neural network sequence models with softmax classifiers have achieved their best language modeling performance only with very large hidden states and large vocabularies. Even then they struggle to predict rare or unseen words even if the context makes the prediction unambiguous. We introduce the pointer sentinel mixture architecture for neural sequence models which has the ability to either reproduce a word from the recent context or produce a word from a standard softmax classifier. We explore applying the pointer sentinel mixture model to the LSTM, a standard recurrent neural network building block. Utilizing an LSTM that achieves 80.6 perplexity on the Penn Treebank, the pointer sentinel-LSTM model pushes perplexity down to 70.9 while using far fewer parameters than an LSTM that achieves similar results. In order to evaluate how well language models can exploit longer contexts and deal with more realistic vocabularies and corpora we also introduce the freely available WikiText corpus.[1]

## 1 INTRODUCTION

A major difficulty in language modeling is learning when to predict specific words from the immediate context. For instance, imagine a new person is introduced and two paragraphs later the context would allow one to very accurately predict this person's name as the next word. For standard neural sequence models to predict this name, they would have to encode the name, store it for many time steps in their hidden state, and then decode it when appropriate. As the hidden state is limited in capacity and the optimization of such models suffer from the vanishing gradient problem, this is a lossy operation when performed over many timesteps. This is especially true for rare words.

Models with soft attention or memory components have been proposed to help deal with this challenge, aiming to allow for the retrieval and use of relevant previous hidden states, in effect increasing hidden state capacity and providing a path for gradients not tied to timesteps. Even with attention, the standard softmax classifier that is being used in these models often struggles to correctly predict rare or previously unknown words.

Pointer networks (Vinyals et al., 2015) provide one potential solution for rare and out of vocabulary (OoV) words as a pointer network uses attention to select an element from the input as output. This allows it to produce previously unseen input tokens. While pointer networks improve performance on rare words and long-term dependencies they are unable to select words that do not exist in the input.

We introduce a mixture model, illustrated in Fig. 1, that combines the advantages of standard softmax classifiers with those of a pointer component for effective and efficient language modeling. Rather than relying on the RNN hidden state to decide when to use the pointer, as in the recent work of Gülçehre et al. (2016), we allow the pointer component itself to decide when to use the softmax vocabulary through a sentinel. The model improves the state of the art perplexity on the Penn Treebank. Since this commonly used dataset is small and no other freely available alternative exists that allows for learning long range dependencies, we also introduce a new benchmark dataset for language modeling called WikiText.

---

[1]Available for download at the WikiText dataset site

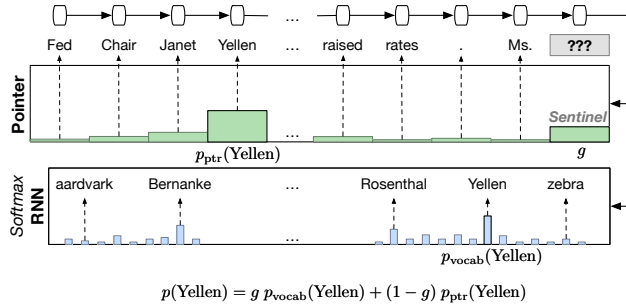

$$p(\text{Yellen}) = g\, p_{\text{vocab}}(\text{Yellen}) + (1-g)\, p_{\text{ptr}}(\text{Yellen})$$

Figure 1: Illustration of the pointer sentinel-RNN mixture model. $g$ is the mixture gate which uses the sentinel to dictate how much probability mass to give to the vocabulary.

## 2 THE POINTER SENTINEL FOR LANGUAGE MODELING

Given a sequence of words $w_1, \ldots, w_{N-1}$, our task is to predict the next word $w_N$.

### 2.1 THE SOFTMAX-RNN COMPONENT

Recurrent neural networks (RNNs) have seen widespread use for language modeling (Mikolov et al., 2010) due to their ability to, at least in theory, retain long term dependencies. RNNs employ the chain rule to factorize the joint probabilities over a sequence of tokens: $p(w_1, \ldots, w_N) = \prod_{i=1}^{N} p(w_i | w_1, \ldots, w_{i-1})$. More precisely, at each time step $i$, we compute the RNN hidden state $h_i$ according to the previous hidden state $h_{i-1}$ and the input $x_i$ such that $h_i = RNN(x_i, h_{i-1})$. When all the $N-1$ words have been processed by the RNN, the final state $h_{N-1}$ is fed into a softmax layer which computes the probability over a vocabulary of possible words: $p_{\text{vocab}}(w) = \text{softmax}(Uh_{N-1})$, where $p_{\text{vocab}} \in \mathbb{R}^V$, $U \in \mathbb{R}^{V \times H}$, $H$ is the hidden size, and $V$ the vocabulary size. RNNs can suffer from the vanishing gradient problem. The LSTM (Hochreiter & Schmidhuber, 1997) architecture has been proposed to deal with this by updating the hidden state according to a set of gates. Our work focuses on the LSTM but can be applied to any RNN architecture that ends in a vocabulary softmax.

### 2.2 THE POINTER NETWORK COMPONENT

In this section, we propose a modification to pointer networks for language modeling. To predict the next word in the sequence, a pointer network would select the member of the input sequence $p(w_1, \ldots, w_{N-1})$ with the maximal attention score as the output.

The simplest way to compute an attention score for a specific hidden state is an inner product with all the past hidden states $h$, with each hidden state $h_i \in \mathbb{R}^H$. However, if we want to compute such a score for the most recent word (since this word may be repeated), we need to include the last hidden state itself in this inner product. Taking the inner product of a vector with itself results in the vector's magnitude squared, meaning the attention scores would be strongly biased towards the most recent word. Hence we project the current hidden state to a query vector $q$ first. To produce the query $q$ we compute $q = \tanh(Wh_{N-1} + b)$, where $W \in \mathbb{R}^{H \times H}$, $b \in \mathbb{R}^H$, and $q \in \mathbb{R}^H$. To generate the pointer attention scores, we compute the match between the previous RNN output states $h_i$ and the query $q$ by taking the inner product, followed by a softmax activation function to obtain a probability distribution:

$$z_i = q^T h_i, \tag{1}$$
$$a = \text{softmax}(z), \tag{2}$$

where $z \in \mathbb{R}^L$, $a \in \mathbb{R}^L$, and $L$ is the total number of hidden states. The probability mass assigned to a given word is the sum of the probability mass given to all token positions where the given word appears:

$$p_{\text{ptr}}(w) = \sum_{i \in I(w,x)} a_i, \tag{3}$$

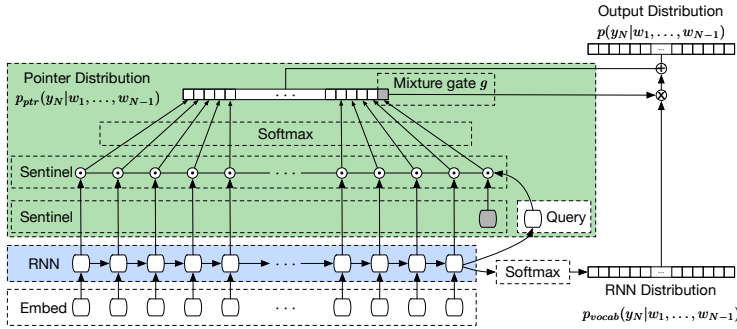

Figure 2: Visualization of the pointer sentinel-RNN mixture model. The query, produced from applying an MLP to the last output of the RNN, is used by the pointer network to identify likely matching words from the past. The $\odot$ nodes are inner products between the query and the RNN hidden states. If the pointer component is not confident, probability mass can be directed to the RNN by increasing the value of the mixture gate $g$ via the sentinel, seen in grey. If $g = 1$ then only the RNN is used. If $g = 0$ then only the pointer is used.

where $I(w, x)$ results in all positions of the word $w$ in the input $x$ and $p_{\text{ptr}} \in \mathbb{R}^V$. This technique, referred to as pointer sum attention, has been used for question answering (Kadlec et al., 2016).

Given the length of the documents used in language modeling, it may not be feasible for the pointer network to evaluate an attention score for all the words back to the beginning of the dataset. Instead, we may elect to maintain only a window of the $L$ most recent words for the pointer to match against. The length $L$ of the window is a hyperparameter that can be tuned on a held out dataset or by empirically analyzing how frequently a word at position $t$ appears within the last $L$ words.

To illustrate the advantages of this approach, consider a long article featuring two sentences *President Obama discussed the economy* and *President Obama then flew to Prague*. If the query was *Which President is the article about?*, probability mass could be applied to *Obama* in either sentence. If the question was instead *Who flew to Prague?*, only the latter occurrence of *Obama* provides the proper context. The attention sum model ensures that, as long as the entire attention probability mass is distributed on the occurrences of *Obama*, the pointer network can achieve zero loss. This flexibility provides supervision without forcing the model to put mass on supervision signals that may be incorrect or lack proper context.

## 2.3 THE POINTER SENTINEL MIXTURE MODEL

While pointer networks have proven to be effective, they cannot predict output words that are not present in the input, a common scenario in language modeling. We propose to resolve this by using a mixture model that combines a standard $\text{softmax}$ with a pointer.

Our mixture model has two base distributions: the $\text{softmax}$ vocabulary of the RNN output and the positional vocabulary of the pointer model. We refer to these as the RNN component and the pointer component respectively. To combine the two base distributions, we use a gating function $g = p(z_i = k | x_i)$ where $z_i$ is the latent variable stating which base distribution the data point belongs to. As we only have two base distributions, $g$ can produce a scalar in the range $[0, 1]$. A value of 0 implies that only the pointer is used and 1 means only the $\text{softmax}$-RNN is used.

$$p(y_i | x_i) = g \, p_{\text{vocab}}(y_i | x_i) + (1 - g) \, p_{\text{ptr}}(y_i | x_i). \tag{4}$$

While the models could be entirely separate, we re-use many of the parameters for the $\text{softmax}$-RNN and pointer components. This sharing minimizes the total number of parameters in the model and capitalizes on the pointer network's supervision for the RNN component.

## 2.4 DETAILS OF THE GATING FUNCTION

To compute the new pointer sentinel gate $g$, we modify the pointer component. In particular, we add an additional element to $z$, the vector of attention scores as defined in Eq. 1. This element is

computed using an inner product between the query and the sentinel[2] vector $s \in \mathbb{R}^H$. This change can be summarized by changing Eq. 2 to $a = \text{softmax}\left(\left[z; q^T s\right]\right)$. We define $a \in \mathbb{R}^{V+1}$ to be the attention distribution over both the words in the pointer window as well as the sentinel state. We interpret the last element of this vector to be the gate value: $g = a[V + 1]$.

Any probability mass assigned to $g$ is given to the standard softmax vocabulary of the RNN. The final updated, normalized pointer probability over the vocabulary in the window then becomes:

$$p_{\text{ptr}}(y_i|x_i) = \frac{1}{1 - g}\ a[1 : V],$$
(5)

where we denoted $[1 : V]$ to mean the first $V$ elements of the vector. The final mixture model is the same as Eq. 4 but with the updated Eq. 5 for the pointer probability.

This setup encourages the model to have both components compete: use pointers whenever possible and back-off to the standard softmax otherwise. By integrating the gating function into the pointer computation, it is influenced by both the RNN hidden state and the pointer window's hidden states.

## 2.5    MOTIVATION FOR THE SENTINEL AS GATING FUNCTION

To make the best decision possible regarding which component to use the gating function must have as much context as possible. As we increase the window of words for the pointer component to consider, the RNN hidden state by itself isn't guaranteed to accurately recall the identity or order of words it has recently seen (Adi et al., 2016). This is an obvious limitation of encoding a variable length sequence into a fixed dimensionality vector.

If we want a pointer window where the length $L$ is in the hundreds, accurately modeling all of this information within the RNN hidden state is impractical. The position of specific words is also a vital feature as relevant words eventually fall out of the pointer component's window. To correctly model this would require the RNN hidden state to store both the identity and position of each word in the pointer window. This is far beyond the capability of the fixed dimensionality RNN hidden state.

For this reason, we integrate the gating function directly into the pointer network by use of the sentinel. The decision to back-off to the softmax vocabulary is then informed by both the query $q$, generated using the RNN hidden state $h_{N-1}$, and from the contents of the hidden states in the pointer window itself. This allows the model to accurately query what hidden states are contained in the pointer window and avoid maintaining state for words that may have fallen out of the window.

## 2.6    POINTER SENTINEL LOSS FUNCTION

We minimize the cross-entropy loss of $-\sum_j \hat{y}_{ij} \log p(y_{ij}|x_i)$, where $\hat{y}_i$ is a one hot encoding of the correct output. During training, as $\hat{y}_i$ is one hot, only a single mixed probability $p(y_{ij})$ must be computed for calculating the loss. This can result in a far more efficient GPU implementation. At prediction time, when we want all values for $p(y_i|x_i)$, a maximum of $L$ word probabilities must be mixed, as there is a maximum of $L$ unique words in the pointer window of length $L$. This mixing can occur on the CPU where random access indexing is more efficient than the GPU.

Following the pointer sum attention network, the aim is to place probability mass from the attention mechanism on the correct output $\hat{y}_i$ if it exists in the input. In the case of our mixture model the pointer loss instead becomes $-\log\left(g + \sum_{i \in I(y,x)} a_i\right)$, where $I(y, x)$ results in all positions of the correct output $y$ in the input $x$. The gate $g$ may be assigned all probability mass if, for instance, the correct output $\hat{y}_i$ exists only in the softmax-RNN vocabulary. There is no penalty if the model places the entire probability mass on any of the instances of the correct word in the input window. If the pointer component places the entirety of the probability mass on the gate $g$, the pointer network incurs no penalty and the loss is entirely determined by the loss of the softmax-RNN component.

---

[2]A sentinel value is inserted at the end of a search space in order to ensure a search algorithm terminates if no matching item is found. Our sentinel value terminates the pointer search space and distributes the rest of the probability mass to the RNN vocabulary.

|  | **Penn Treebank** | | | **WikiText-2** | | | **WikiText-103** | | |
|  | Train | Valid | Test | Train | Valid | Test | Train | Valid | Test |
|---|---|---|---|---|---|---|---|---|---|
| Articles | - | - | - | 600 | 60 | 60 | 28,475 | 60 | 60 |
| Tokens | 929k | 73k | 82k | 2,088k | 217k | 245k | 103,227k | 217k | 245k |
| Vocab size | 10,000 | | | 33,278 | | | 267,735 | | |
| OoV rate | 4.8% | | | 2.6% | | | 0.4% | | |

Table 1: Statistics of the Penn Treebank, WikiText-2, and WikiText-103. The out of vocabulary (OoV) rate notes what percentage of tokens have been replaced by an $\langle unk \rangle$ token. The token count includes newlines which add to the structure of the WikiText datasets.

## 2.7 PARAMETERS AND COMPUTATION TIME

The pointer sentinel-LSTM mixture model results in a relatively minor increase in parameters and computation time, especially when compared to the model size required to achieve similar performance using a standard LSTM. The only two additional parameters required by the model are those required for computing $q$, specifically $W \in \mathbb{R}^{H \times H}$ and $b \in \mathbb{R}^H$, and the sentinel vector embedding, $s \in \mathbb{R}^H$. This is independent of the depth of the RNN as the pointer component only interacts with the output of the final RNN layer. The additional $H^2 + 2H$ parameters are minor compared to a single LSTM layer's $8H^2 + 4H$ parameters. Most models also use multiple LSTM layers.

In terms of additional computation, a pointer sentinel-LSTM of window size $L$ only requires computing the query $q$ (a linear layer with $\tanh$ activation), a total of $L$ parallelizable inner product calculations, and the attention scores for the $L$ resulting scalars via the $\mathrm{softmax}$ function.

## 3 RELATED WORK

Considerable research has been dedicated to the task of language modeling, from traditional machine learning techniques such as n-grams to deep neural sequence models.

Mixture models composed of various knowledge sources have been proposed in the past for language modeling. Rosenfeld (1996) uses a maximum entropy model to combine a variety of information sources to improve language modeling on news text and speech. These information sources include complex overlapping n-gram distributions and n-gram caches that aim to capture rare words.

Beyond n-grams, neural sequence models such as recurrent neural networks have been shown to achieve state of the art results (Mikolov et al., 2010). A variety of RNN regularization methods have been explored, including a number of dropout variations (Zaremba et al., 2014; Gal, 2015) which prevent overfitting of complex LSTM language models. Other work has modified the RNN architecture to better handle increased recurrence depth (Zilly et al., 2016).

In order to increase capacity and minimize the impact of vanishing gradients, some language and translation models have also added a soft attention or memory component (Bahdanau et al., 2015; Sukhbaatar et al., 2015; Cheng et al., 2016; Kumar et al., 2016; Xiong et al., 2016; Ahn et al., 2016). These mechanisms allow for the retrieval and use of relevant previous hidden states. Soft attention mechanisms need to first encode the relevant word into a state vector and then decode it again, even if the output word is identical to the input word used to compute that hidden state or memory. A drawback to soft attention is that if, for instance, *January* and *March* are both equally attended candidates, the attention mechanism may blend the two vectors, resulting in a context vector closest to *February* (Kadlec et al., 2016). Even with attention, the standard $\mathrm{softmax}$ classifier being used in these models often struggles to correctly predict rare or previously unknown words.

Attention-based pointer mechanisms were introduced in Vinyals et al. (2015) where the pointer network is able to select elements from the input as output. In the above example, only *January* or *March* would be available as options, as *February* does not appear in the input. The use of pointer networks have been shown to help with geometric problems (Vinyals et al., 2015), code generation (Ling et al., 2016), summarization (Gu et al., 2016; Gülçehre et al., 2016), question answering (Kadlec et al., 2016). While pointer networks improve performance on rare words and long-term dependencies they are unable to select words that do not exist in the input.

Gülçehre et al. (2016) introduce a pointer softmax model that can generate output from either the vocabulary softmax of an RNN or the location softmax of the pointer network. Not only does this allow for producing OoV words which are not in the input, the pointer softmax model is able to better deal with rare and unknown words than a model only featuring an RNN softmax. Rather than constructing a mixture model as in our work, they use a switching network to decide which component to use. For neural machine translation, the switching network is conditioned on the representation of the context of the source text and the hidden state of the decoder. The pointer network is not used as a source of information for the switching network as in our model. The pointer and RNN softmax are scaled according to the switching network and the word or location with the highest final attention score is selected for output. Although this approach uses both a pointer and RNN component, it is not a mixture model and does not combine the probabilities for a word if it occurs in both the pointer location softmax and the RNN vocabulary softmax. In our model the word probability is a mix of both the RNN and pointer components, allowing for better predictions when the context may be ambiguous.

Extending this concept further, the latent predictor network (Ling et al., 2016) generates an output sequence conditioned on an arbitrary number of base models where each base model may have differing granularity. In their task of code generation, the output could be produced one character at a time using a standard softmax or instead copy entire words from referenced text fields using a pointer network. As opposed to Gülçehre et al. (2016), all states which produce the same output are merged by summing their probabilities. The model requires a complex training process involving the forward-backward algorithm for Semi-Markov models to prevent an exponential path explosion.

# 4 WikiText - A Benchmark for Language Modeling

We first describe the most commonly used language modeling dataset and its pre-processing in order to then motivate the need for a new benchmark dataset.

## 4.1 Penn Treebank

In order to compare our model to the many recent neural language models, we conduct word-level prediction experiments on the Penn Treebank (PTB) dataset (Marcus et al., 1993), pre-processed by Mikolov et al. (2010). The dataset consists of 929k training, 73k validation, and 82k test words. As part of the pre-processing performed by Mikolov et al. (2010), words were lower-cased, numbers were replaced with N, newlines were replaced with $\langle eos \rangle$, and all other punctuation was removed. The vocabulary is the most frequent 10k words with OoV tokens replaced by an $\langle unk \rangle$ token. For full statistics, refer to Table 1.

## 4.2 Reasons for a New Dataset

While the processed version of the PTB above has been frequently used for language modeling, it has many limitations. The tokens in PTB are all lower case, stripped of any punctuation, and limited to a vocabulary of only 10k words. These limitations mean that the PTB is unrealistic for real language use, especially when far larger vocabularies with many rare words are involved. The appendix contains a graph illustrating this using a Zipfian plot over the training partition of the PTB, with the curve stopping abruptly at the 10k limit. Given that accurately predicting rare words, such as named entities, is an important task for many applications, the lack of a long tail is problematic.

Other larger scale language modeling datasets exist. Unfortunately, they either have restrictive licensing which prevents widespread use or have randomized sentence ordering (Chelba et al., 2013) which is unrealistic for most language use and prevents the effective learning and evaluation of longer term dependencies. Hence, we constructed a language modeling dataset using text extracted from Wikipedia and have made this available to the community.

## 4.3 Construction and Pre-processing

We selected articles only fitting the *Good* or *Featured* article criteria specified by editors on Wikipedia. These articles have been reviewed by humans and are considered well written, factually accurate, broad in coverage, neutral in point of view, and stable. This resulted in 23,805 Good

articles and 4,790 Featured articles. The text for each article was extracted using the Wikipedia API. Extracting text from Wikipedia mark-up is nontrivial due to the large number of macros in use, used for metric conversions, abbreviations, language notation, and date handling.

Once extracted, specific sections which primarily featured lists were removed by default. Other minor bugs, such as sort keys and Edit buttons that leaked in from the HTML, were also removed. Mathematical formulae and LaTeX code were replaced with $\langle formula \rangle$ tokens. Normalization and tokenization were performed using the Moses tokenizer (Koehn et al., 2007), slightly augmented to further split numbers ($8,600 \rightarrow 8$ @,@ 600) and with some additional minor fixes. Following Chelba et al. (2013) a vocabulary was constructed by discarding all words with a count below 3. Words outside of the vocabulary were mapped to the $\langle unk \rangle$ token, also a part of the vocabulary.

## 4.4 STATISTICS

The full WikiText dataset is over 103 million words in size, a hundred times larger than the PTB. It is also a tenth the size of the One Billion Word Benchmark (Chelba et al., 2013), one of the largest publicly available language modeling benchmarks, whilst consisting of articles that allow for the capture and usage of longer term dependencies as might be found in many real world tasks.

The dataset is available in two different sizes: WikiText-2 and WikiText-103. Both feature punctuation, original casing, a larger vocabulary, and numbers. WikiText-2 is two times the size of the Penn Treebank dataset. WikiText-103 features all extracted articles. Both datasets use the same articles for validation and testing, only differing in the vocabularies. For full statistics, refer to Table 1.

## 5 EXPERIMENTS

### 5.1 TRAINING DETAILS

As the pointer sentinel mixture model uses the outputs of the RNN from up to $L$ timesteps back, this presents a challenge for training. If we do not regenerate the stale historical outputs of the RNN when we update the gradients, backpropagation through these stale outputs may result in incorrect gradient updates. If we do regenerate all stale outputs of the RNN, the training process is far slower. As we can make no theoretical guarantees on the impact of stale outputs on gradient updates, we opt to regenerate the window of RNN outputs used by the pointer component after each gradient update.

We also use truncated backpropagation through time (BPTT) in a different manner to many other RNN language models. Truncated BPTT allows for practical time-efficient training of RNN models but has fundamental trade-offs that are rarely discussed. For running truncated BPTT, BPTT is run for $k_2$ timesteps once every $k_1$ timesteps. For many RNN language modeling training schemes, $k_1 = k_2$, meaning that every $k$ timesteps truncated BPTT is performed for the $k$ previous timesteps. This results in only a single RNN output receiving backpropagation for $k$ timesteps, with the other extreme being that the first token receives backpropagation for 0 timesteps. As such, most words in the training data will never experience a full backpropagation for $k$ timesteps.

In our task, the pointer component always looks $L$ timesteps into the past if $L$ past timesteps are available. We select $k_1 = 1$ and $k_2 = L$ such that for each timestep we perform backpropagation for $L$ timesteps and advance one timestep at a time. Only the loss for the final predicted word is used for backpropagation through the window.

### 5.2 MODEL DETAILS

Our experimental setup reflects that of Zaremba et al. (2014) and Gal (2015). We increased the number of timesteps used during training from 35 to 100, matching the length of the window $L$. Batch size was increased to 32 from 20. We also halve the learning rate when validation perplexity is worse than the previous iteration, stopping training when validation perplexity fails to improve for three epochs or when 64 epochs are reached. The gradients are rescaled if their global norm exceeds 1 (Pascanu et al., 2013b).[3] We evaluate the medium model configuration which features a two layer

---

[3]The highly aggressive clipping is likely due to the increased BPTT length. Even with such clipping early batches may experience excessively high perplexity, though this settles rapidly.

| Model | Parameters | Validation | Test |
|---|---|---|---|
| Mikolov & Zweig (2012) - KN-5 | 2M‡ | – | 141.2 |
| Mikolov & Zweig (2012) - KN5 + cache | 2M‡ | – | 125.7 |
| Mikolov & Zweig (2012) - RNN | 6M‡ | – | 124.7 |
| Mikolov & Zweig (2012) - RNN-LDA | 7M‡ | – | 113.7 |
| Mikolov & Zweig (2012) - RNN-LDA + KN-5 + cache | 9M‡ | – | 92.0 |
| Pascanu et al. (2013a) - Deep RNN | 6M | – | 107.5 |
| Cheng et al. (2014) - Sum-Prod Net | 5M‡ | – | 100.0 |
| Zaremba et al. (2014) - LSTM (medium) | 20M | 86.2 | 82.7 |
| Zaremba et al. (2014) - LSTM (large) | 66M | 82.2 | 78.4 |
| Gal (2015) - Variational LSTM (medium, untied) | 20M | $81.9 \pm 0.2$ | $79.7 \pm 0.1$ |
| Gal (2015) - Variational LSTM (medium, untied, MC) | 20M | – | $78.6 \pm 0.1$ |
| Gal (2015) - Variational LSTM (large, untied) | 66M | $77.9 \pm 0.3$ | $75.2 \pm 0.2$ |
| Gal (2015) - Variational LSTM (large, untied, MC) | 66M | – | $73.4 \pm 0.0$ |
| Kim et al. (2016) - CharCNN | 19M | – | 78.9 |
| Zilly et al. (2016) - Variational RHN | 32M | 72.8 | 71.3 |
| Zoneout + Variational LSTM (medium) | 20M | 84.4 | 80.6 |
| Pointer Sentinel-LSTM (medium) | 21M | 72.4 | **70.9** |

Table 2: Single model perplexity on validation and test sets for the Penn Treebank language modeling task. For our models and the models of Zaremba et al. (2014) and Gal (2015), medium and large refer to a 650 and 1500 unit two layer LSTM respectively. Parameter numbers with ‡ are estimates based upon our understanding of the model and with reference to Kim et al. (2016).

LSTM of hidden size 650. We compare against the large model configuration which features a two layer LSTM of hidden size 1500.

We produce results for two model types, an LSTM model that uses dropout regularization and the pointer sentinel-LSTM model. The variants of dropout used were zoneout (Krueger et al., 2016) and variational inference based dropout (Gal, 2015). Zoneout, which stochastically forces some recurrent units to maintain their previous values, was used for the recurrent connections within the LSTM. Variational inference based dropout, where the dropout mask for a layer is locked across timesteps, was used on the input to each RNN layer and also on the output of the final RNN layer. We used a value of 0.5 for both dropout connections.

## 5.3 COMPARISON OVER PENN TREEBANK

Table 2 compares the pointer sentinel-LSTM to a variety of other models on the Penn Treebank dataset. The pointer sentinel-LSTM achieves the lowest perplexity, followed by the recent Recurrent Highway Networks (Zilly et al., 2016). The medium pointer sentinel-LSTM model also achieves lower perplexity than the large LSTM models. Note that the best performing large variational LSTM model uses computationally intensive Monte Carlo (MC) dropout averaging. Monte Carlo dropout averaging is a general improvement for any sequence model that uses dropout but comes at a greatly increased test time cost. In Gal (2015) it requires rerunning the test model with 1000 different dropout masks. The pointer sentinel-LSTM is able to achieve these results with far fewer parameters than other models with comparable performance, specifically with less than a third the parameters used in the large variational LSTM models.

We also test a variational LSTM that uses zoneout, which serves as the RNN component of our pointer sentinel-LSTM mixture. This variational LSTM model performs BPTT for the same length $L$ as the pointer sentinel-LSTM, where $L = 100$ timesteps. The results for this model ablation are worse than that of Gal (2015)'s variational LSTM without Monte Carlo dropout averaging.

## 5.4 COMPARISON OVER WIKITEXT-2

As WikiText-2 is being introduced in this dataset, there are no existing baselines. We provide two baselines to compare the pointer sentinel-LSTM against: our variational LSTM using zoneout and

| Model | Parameters | Validation | Test |
|---|---|---|---|
| Variational LSTM implementation from Gal (2015) | 20M | 101.7 | 96.3 |
| Zoneout + Variational LSTM | 20M | 108.7 | 100.9 |
| Pointer Sentinel-LSTM | 21M | 84.8 | **80.8** |

Table 3: Single model perplexity on validation and test sets for the WikiText-2 language modeling task. All compared models use a two layer LSTM with a hidden size of 650.

the medium variational LSTM used in Gal (2015).[4] Attempts to run the Gal (2015) large model variant, a two layer LSTM with hidden size 1500, resulted in out of memory errors on a 12GB K80 GPU, likely due to the increased vocabulary size. We chose the best hyperparameters from PTB experiments for all models. Table 3 shows a similar gain made by the pointer sentinel-LSTM over the variational LSTM models. The variational LSTM from Gal (2015) again beats out the variational LSTM used as a base for our experiments.

# 6 ANALYSIS

## 6.1 IMPACT ON RARE WORDS

A hypothesis as to why the pointer sentinel-LSTM can outperform an LSTM is that the pointer component allows the model to effectively reproduce rare words. The RNN may better use hidden state capacity by relying on the pointer component. The pointer component may also allow for a sharper selection of a single word than may be possible using only the softmax.

The appendix contains a graph which shows the improvement of perplexity when comparing the LSTM to the pointer sentinel-LSTM. Words are split across buckets according to frequency. As the words become rarer, the pointer sentinel-LSTM has stronger improvements in perplexity. Even on the Penn Treebank, where there is a relative absence of rare words due to only selecting the most frequent 10k words, we can see the pointer sentinel-LSTM mixture model provides a direct benefit.

While the improvements are largest on rare words, we can see the pointer sentinel-LSTM is still helpful on relatively frequent words. This may be the pointer component directly selecting the word or through the pointer supervision signal improving the RNN by allowing gradients to flow directly to other occurrences of the word in that window.

## 6.2 QUALITATIVE ANALYSIS OF POINTER USAGE

In a qualitative analysis, we visualized the gate use and pointer attention for a variety of examples in the validation set, focusing on predictions where the gate primarily used the pointer component. These visualizations are available in the appendix.

As expected, the pointer component is heavily used for rare names such as *Seidman* (23 times in training), *Iverson* (7 times in training), and *Rosenthal* (3 times in training). The pointer component was also heavily used when it came to other named entity names such as companies like *Honeywell* (8 times in training) and *Integrated* (41 times in training, though due to lowercasing of words this includes *integrated circuits*, *fully integrated*, and other generic usage). Surprisingly, the pointer component was also used for many frequent tokens. For selecting units of measurement (tons, kilograms, . . . ) or the short scale of numbers (thousands, millions, billions, . . . ), the pointer would refer to recent usage. This is to be expected, especially when phrases are of the form *increased from N tons to N tons*. The model can even be found relying on a mixture of the softmax and the pointer for predicting frequent verbs such as *said*.

Finally, the pointer component can be seen pointing to words at the very end of the 100 word window (position 97), a far longer horizon than the 35 steps that most language models truncate their backpropagation training to. This illustrates why the gating function must be integrated into the pointer component. If the gating function could only use the RNN hidden state, it would need to be wary of words that were near the tail of the pointer, especially if it was not able to accurately

---

[4]https://github.com/yaringal/BayesianRNN

track exactly how long it was since seeing a word. By integrating the gating function into the pointer component, we avoid the RNN hidden state having to maintain this intensive bookkeeping.

## 7 CONCLUSION

We introduced the pointer sentinel mixture model and the WikiText language modeling dataset. The pointer sentinel mixture model can be applied to any classifier that ends in a $\mathrm{softmax}$, including various recurrent neural network building blocks. When applied to a standard LSTM, the pointer sentinel-LSTM achieves state of the art results in language modeling over the Penn Treebank while using few additional parameters and little additional computational complexity at prediction time.

We have also motivated the need to move from Penn Treebank to a new language modeling dataset for long range dependencies, providing WikiText-2 and WikiText-103 as potential options. We hope these new datasets can serve as a platform to improve handling of rare words and the usage of long term dependencies in language modeling.

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

APPENDIX

PERPLEXITY IMPROVEMENTS FOR POINTER SENTINEL MIXTURE MODEL

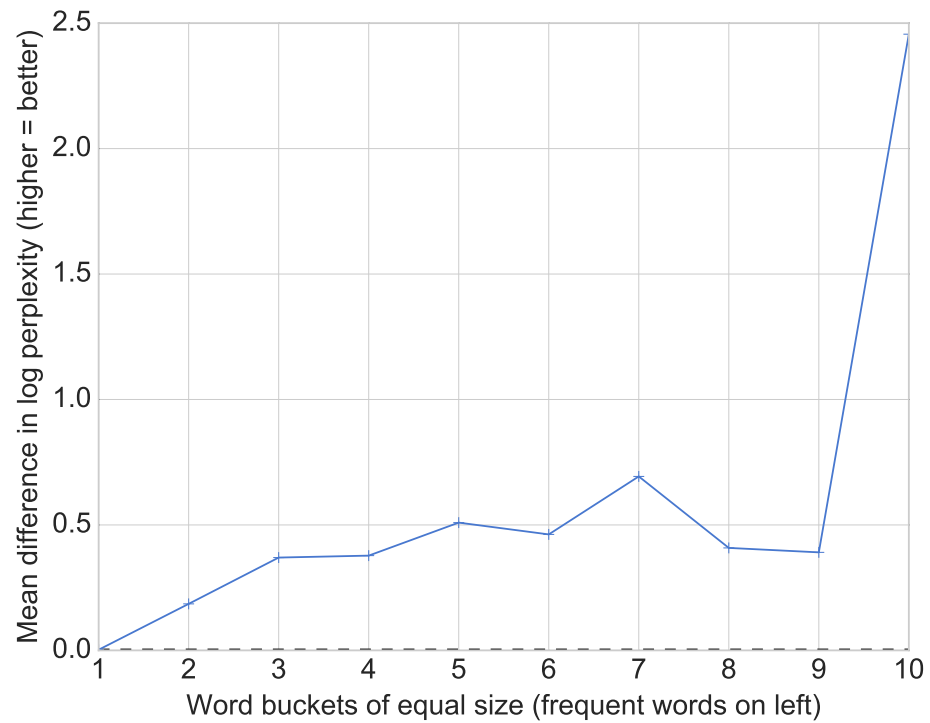

Figure A1: Mean difference in log perplexity on PTB when using the pointer sentinel-LSTM compared to the LSTM model. Words were sorted by frequency and split into equal sized buckets.

POINTER USAGE ON THE PENN TREEBANK

For a qualitative analysis, we visualize how the pointer component is used within the pointer sentinel mixture model. The gate refers to the result of the gating function, with 1 indicating the RNN component is exclusively used whilst 0 indicates the pointer component is exclusively used. We begin with predictions that are using the RNN component primarily and move to ones that use the pointer component primarily.

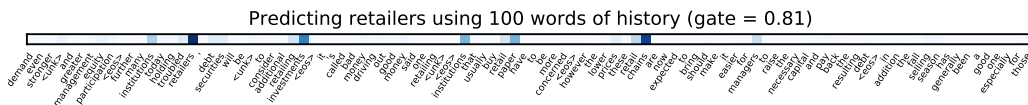

Figure A2: In predicting *the fall season has been a good one especially for those* **retailers**, the pointer component suggests many words from the historical window that would fit - *retailers*, *investments*, *chains*, and *institutions*. The gate is still primarily weighted towards the RNN component however.

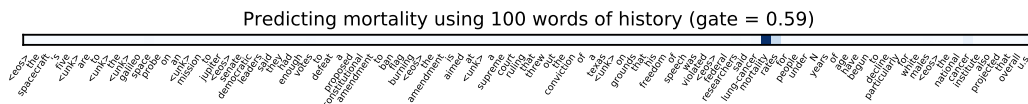

Figure A3: In predicting *the national cancer institute also projected that overall u.s.* **mortality**, the pointer component is focused on *mortality* and *rates*, both of which would fit. The gate is still primarily weighted towards the RNN component.

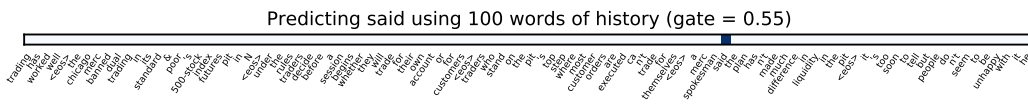

Figure A4: In predicting *people do n't seem to be unhappy with it he* **said**, the pointer component correctly selects *said* and is almost equally weighted with the RNN component. This is surprising given how frequent the word **said** is used within the Penn Treebank.

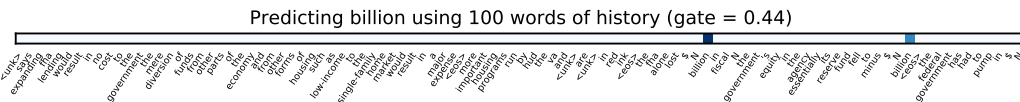

Figure A5: For predicting *the federal government has had to pump in $ N* **billion**, the pointer component focuses on the recent usage of billion with highly similar context. The pointer component is also relied upon more heavily than the RNN component - surprising given the frequency of **billion** within the Penn Treebank and that the usage was quite recent.

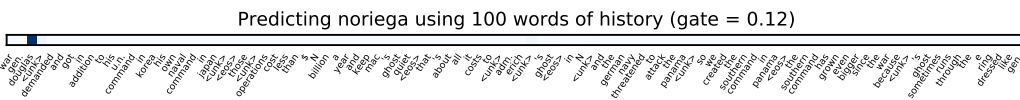

Figure A6: For predicting ⟨*unk*⟩ *'s ghost sometimes runs through the e ring dressed like gen.* **noriega**, the pointer component reaches 97 timesteps back to retrieve gen. douglas. Unfortunately this prediction is incorrect but without additional context a human would have guessed the same word. This additionally illustrates why the gating function must be integrated into the pointer component. The named entity *gen. douglas* would have fallen out of the window in only four more timesteps, a fact that the RNN hidden state would not be able to accurately retain for almost 100 timesteps.

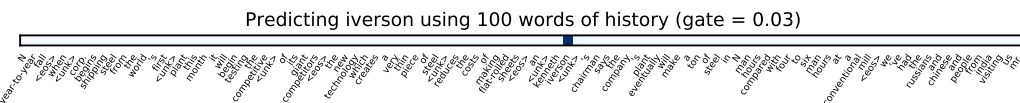

Figure A7: For predicting *mr.* **iverson**, the pointer component has learned the ability to point to the last name of the most recent named entity. The named entity also occurs 45 timesteps ago, which is longer than the 35 steps that most language models truncate their backpropagation to.

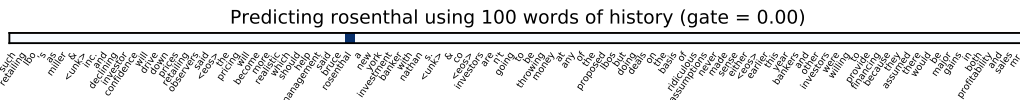

Figure A8: For predicting *mr.* **rosenthal**, the pointer is almost exclusively used and reaches back 65 timesteps to identify *bruce rosenthal* as the person speaking, correctly only selecting the last name.

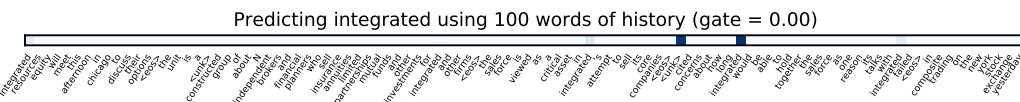

Figure A9: For predicting *in composite trading on the new york stock exchange yesterday* **integrated**, the company Integrated and the ⟨*unk*⟩ token are primarily attended to by the pointer component, with nearly the full prediction being determined by the pointer component.

ZIPFIAN PLOT OVER PTB AND WIKITEXT-2

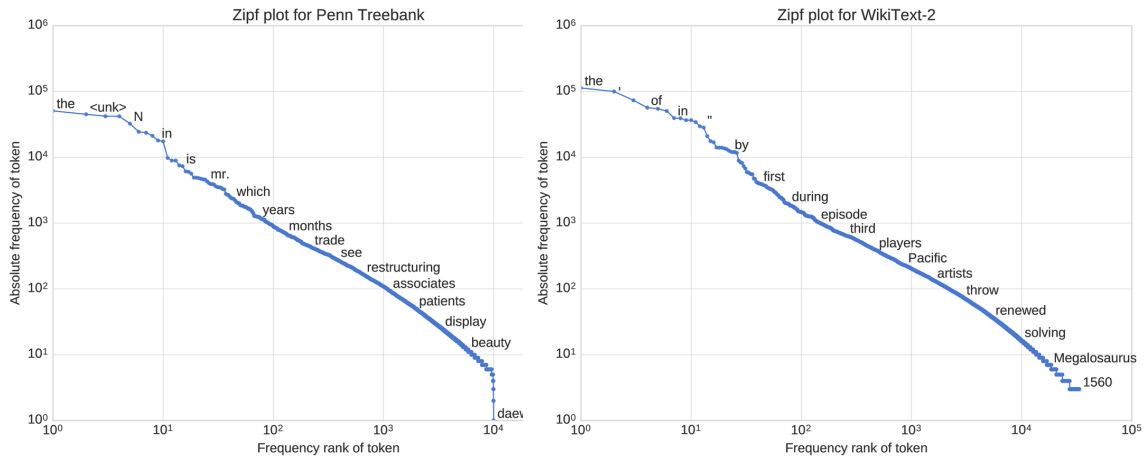

Figure A10: Zipfian plot over the training partition in Penn Treebank and WikiText-2 datasets. Notice the severe drop on the Penn Treebank when the vocabulary hits $10^4$. Two thirds of the vocabulary in WikiText-2 are past the vocabulary cut-off of the Penn Treebank.

ZIPFIAN PLOT OVER WIKITEXT-103

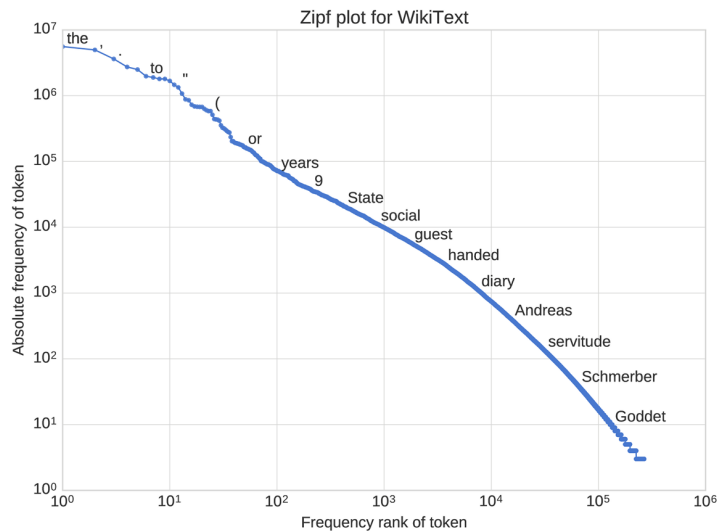

Figure A11: Zipfian plot over the training partition in the WikiText-103 dataset. With the dataset containing over 100 million tokens, there is reasonable coverage of the long tail of the vocabulary.

