# Peer review of "Pointer Sentinel Mixture Models"

_ICLR 2017 — accepted_

[Official Review · AnonReviewer3 · rating 8 · confidence 4 · 16 Dec 2016]

This work is basically a combined pointer network applied on language modelling. 
The smart point is that this paper aims at language modelling with longer context, where a memory of seen words (especially the rare words) would be very useful for predicting the rest of the sentences. 
Hence, a combination of a pointer network and a standard language model would balance the copying seen words and predicting unseen words. 

Generally, such as the combined pointer networks applied in sentence compression, a vector representation of the source sequence would be used to compute the gate. 
This paper, instead, introduces a sentinel vector to carry out the mixture model, which is suitable in the case of language modelling. 
I would be interested in the variations of sentinel mixture implementation, though the current version has achieved very good results. 

In addition, the new WikiText language modelling dataset is very interesting. 
It probably can be a more standard dataset for evaluating the continuously-updated language model benchmarks than ptb dataset. 

Overall, this is a well-written paper. I recommend it to be accepted.

[Official Review · AnonReviewer2 · rating 8 · confidence 4 · 16 Dec 2016]
**A nice approach for rare words / context biasing for LM**

This work is an extension of previous works on pointer models, that mixes its outputs with standard softmax outputs. 
The idea is appealing in general for context biasing and the specific approach appears quite simple.

The idea is novel to some extent, as previous paper had already tried to combine pointer-based and standard models,
but not as a mixture model, as in this paper.

The paper is clearly written and the results seem promising.
The new dataset the authors created (WikiText) also seems of high interest. 

A comment regarding notation:
The symbol p_ptr is used in two different ways in eq. 3 and eq. 5. : p_ptr(w) vs. p_ptr(y_i|x_i) 
This is confusing as these are two different domains: for eq 3. the domain is a *set* of words and for eq. 5 the domain is a *list* of context words.
It would be helpful to use different symbol for the two objects.

[Official Review · AnonReviewer1 · rating 7 · confidence 4 · 27 Dec 2016 (modified: 20 Jan 2017)]

This paper proposes augmenting RNN-based language models with a pointer network in order to deal better with rare words. The pointer network can point to words in the recent context, and hence the prediction for each time step is a mixture between the usual softmax output and the pointer distribution over the recent words. The paper also introduces a new language modelling dataset, which overcomes some of the shortcomings of previous datasets.

The reason for the score I gave for this paper is that I find the proposed model a direct application of the previous work Gulcehre et al., which follows a similar approach but for machine translation and summarization. The main differences I find is that Gulcehre et al. use an encoder-decoder architecture, and use the attention weights of the encoder to point to locations of words in the input, while here an RNN is used and a pointer network produces a distribution over the full vocabulary (by summing the softmax probabilities of words in the recent context). The context (query) vector for the pointing network is also different, but this is also a direct consequence of having a different application.

While the paper describes the differences between the proposed approach and Gulcehre et al.’s approach, I find some of the claims either wrong or not that significant. For example, quoting from Section 1:
“Rather than relying on the RNN hidden state to decide when to use the pointer, as in the recent work of Gulcehre et al. (2016), we allow the pointer component itself to decide when to use the softmax vocabulary through a sentinel.”
As far as I can tell, your model also uses the recent hidden state to form a query vector,  which is matched by the pointer network to previous words. Can you please clarify what you mean here?

In addition, quoting from section 3 which describes the model of Gulcehre et al.:
“Rather than constructing a mixture model as in our work, they use a switching network to decide which component to use”
This is not correct. The model of Gulcehre is also a mixture model, where an MLP with sigmoid output (switching network) is used to form a mixture between softmax prediction and locations of the input text.

Finally, in the following quote, also from section 3: 
“The pointer network is not used as a source of information for the switching network as in our model.” 
It is not clear what the authors mean by “source of information” here. Is it the fact that the switching probability is part of the pointer softmax? I am wondering how significant this difference is.

With regards to the proposed dataset, there are also other datasets typically used for language modelling, including The Hutter Prize Wikipedia (enwik8) dataset (Hutter, 2012) and e Text8 dataset (Mahoney, 2009). Can you please comment on the differences between your dataset and those as well?

I would be happy to discuss with the authors the points I raised, and I am open to changing my vote if there is any misunderstanding on my part.

[Final Decision · Program Chairs · 06 Feb 2017]
**ICLR committee final decision**

The reviewers liked this paper quite a bit, and so for this reason it is a perfectly fine paper to accept. However, it should be noted that the area chair was less enthusiastic. The area chairs mentions that the model appears to be an extension of Gulcehre et al. and the Penn Treebank perplexity experiments are too small scale to be taken seriously in 2017. Instead of experimenting on other known large-scale language modeling setups, the authors introduce their own new dataset (which is 1 order of magnitude smaller than the 1-Billion LM dataset by Chelba et al). The new dataset might be a good idea, but the area chair doesn't understand why the authors do not run public available systems as baselines. This should have been fairly easy to do and would have significantly strengthen the result of this work. The PCs thus encourage to authors to take into account this feedback and consider updating their paper accordingly.